# The Interplay between Cancer Biology and the Endocannabinoid System—Significance for Cancer Risk, Prognosis and Response to Treatment

**DOI:** 10.3390/cancers12113275

**Published:** 2020-11-05

**Authors:** Estefanía Moreno, Milena Cavic, Ana Krivokuca, Enric I. Canela

**Affiliations:** 1Department of Biochemistry and Molecular Biomedicine, Faculty of Biology, University of Barcelona, 08028 Barcelona, Spain; 2Institute of Biomedicine of the University of Barcelona (IBUB), 08028 Barcelona, Spain; 3Department of Experimental Oncology, Institute for Oncology and Radiology of Serbia, Pasterova 14, 11000 Belgrade, Serbia; milena.cavic@ncrc.ac.rs (M.C.); krivokuca.ana@gmail.com (A.K.); 4Centro de Investigación Biomédica en Red sobre Enfermedades Neurodegenerativas (CIBERNED), 28031, Madrid, Spain

**Keywords:** anti-cancer treatment, cancer risk, cannabinoid receptors, cannabinoids

## Abstract

**Simple Summary:**

This review analyses the complex involvement of the various components of the endocannabinoid system (ECS) in the susceptibility to cancer, prognosis, and response to treatment, focusing on its relationship with cancer biology in selected solid cancers (breast, gastrointestinal, gynaecological, prostate cancer, thoracic, thyroid, central nervous system (CNS) tumours, and melanoma). The same ECS component can exert both protective and pathogenic effects in different tumour subtypes, which are often pathologically driven by different biological factors. Although an attractive target in cancer, the use of components in anti-cancer treatment is still interlinked with many legal and ethical issues that need to be considered. The legislation which outlines the permissive boundaries of their therapeutic use in oncology is still unable to follow the current scientific burden of evidence, but the number of ongoing clinical trials might tip the scale forward in the near future.

**Abstract:**

The various components of the endocannabinoid system (ECS), such as the cannabinoid receptors (CBRs), cannabinoid ligands, and the signalling network behind it, are implicated in several tumour-related states, both as favourable and unfavourable factors. This review analyses the ECS’s complex involvement in the susceptibility to cancer, prognosis, and response to treatment, focusing on its relationship with cancer biology in selected solid cancers (breast, gastrointestinal, gynaecological, prostate cancer, thoracic, thyroid, CNS tumours, and melanoma). Changes in the expression and activation of CBRs, as well as their ability to form distinct functional heteromers affect the cell’s tumourigenic potential and their signalling properties, leading to pharmacologically different outcomes. Thus, the same ECS component can exert both protective and pathogenic effects in different tumour subtypes, which are often pathologically driven by different biological factors. The use of endogenous and exogenous cannabinoids as anti-cancer agents, and the range of effects they might induce (cell death, regulation of angiogenesis, and invasion or anticancer immunity), depend in great deal on the tumour type and the specific ECS component that they target. Although an attractive target, the use of ECS components in anti-cancer treatment is still interlinked with many legal and ethical issues that need to be considered.

## 1. Introduction

Historically, cannabinoids have primarily been used as palliative care agents in oncology. However, the various components of the endocannabinoid system (ECS), such as the cannabinoid receptors (CBRs), cannabinoid ligands, and their signalling network are interlinked with several tumour-related states, both as favourable and unfavourable factors. This vast network of molecules is an attractive pharmacological target, and its full potential is yet to be reached. Understanding the specific ways ECS components can regulate the cell cycle, proliferation and cell death considering their interactions with the immune system is necessary for advancing the current state of the art in cannabinoid-based anti-cancer therapeutic approaches. This review analyses the ECS’s complex involvement in the susceptibility to cancer, prognosis and response to treatment, focusing on its relationship with cancer biology in selected solid cancers (breast, gastrointestinal, gynaecological, prostate cancer, thoracic, thyroid, CNS tumours, and melanoma).

## 2. The Interplay between Cancer Biology and the Endocannabinoid System

So far, seven receptors that respond to endogenous and exogenous cannabinoid ligands in humans have been described in literature [1], namely the main cannabinoid receptors 1 and 2 (CB1R, coded by *CNR1* gene [2] and CB2R, coded by *CNR2* gene [3]), as well as G protein-coupled receptors 18 (N-arachidonyl glycine receptor, GPR18 [4]), 55 (GPR55 [5]) and 119 (Glucose-dependent insulinotropic receptor, GPR119 [6]) and the transient receptor potential cation channel subfamily V members 1 and 2 (TRPV1 [7], TRPV2 [8]). All these receptors might be useful anti-cancer targets individually, as well as in various heteromerization scenarios. A simple STRING analysis [9], showed that cannabinoid receptors CB1R and CB2R directly interact with each other using several evidence platforms, as well with GPR18, GPR55 and TRPV1 (Figure 1a). Additional cluster analysis extended to five primary-interaction shell genes showed that GPR119 is only indirectly connected with the other receptors while TRPV2 seems to form a separate network entity (Figure 1b). The extended network is enriched in interactions (PPI enrichment *p*-value: 2.39 × 10^−12^), meaning that these seven receptors in general interact more than is expected for a random set of molecules of similar size and can be considered, at least partially, as a biologically connected group [9].

Changes in expression and activation of these CBRs, as well as their ability to form distinct functional heteromers with many other receptors alter the cell’s tumuorigenic potential and their signalling properties, leading to pharmacologically different outcomes upon their stimulation [10,11,12]. Thus, the same ECS component can exert both protective and pathogenic effects in different tumour subtypes, which are often pathologically driven by different biological factors.

Cannabinoid receptors are widely expressed on normal and cancer cells. The interactive open-access databases the Human Protein Atlas [13,14] and UALCAN [15,16] were used to analyse the Cancer Genome Atlas (TCGA) [17] transcriptome data and assess their expression in various cancer subtypes. In silico analyses showed that cannabinoid receptors were generally not prognostically significant, but are enriched (mostly at the RNA level where more data is available) in some cancer subtypes: CNR1 in glioma (Figure 2a), CNR2 in testicular cancer (Figure 2b), GPR119 in pancreatic cancer, where it is also a favourable prognostic factor (*p* < 0.001, Figure 2c [18]) and TRPV2 in melanoma, while it is an unfavourable prognostic factor in renal (*p* < 0.001, Figure 2d [19]) and testicular cancer (*p* < 0.001, Figure 2e [20]).

The receptors are also significantly over- or under-expressed in some cancer subtypes compared to normal tissue which might be useful for diagnostics and specific anti-cancer approaches (Table 1) [16]. Interestingly, all cannabinoid receptors were found to be significantly under-expressed in colon adenocarcinoma.

### 2.1. Breast Cancer

Breast cancer (BC) remains the most common malignant disease in women in Western countries. Although the rates of mortality have declined since the late 1990s primarily due to adjuvant systemic therapy and earlier detection by palpation and mammograms, some breast tumours remain resistant to conventional therapies. In addition, actual treatments have side effects that substantially affect the patients’ quality of life [21,22,23,24] and many plants have been evaluated as supplementary and alternative anticancer medicines [25,26]. As BC is a highly heterogeneous disease in terms of molecular portraits, prognosis, and treatment [23], there are three main BC subtypes based on classical molecular profiles: hormone receptor-positive, Human epidermal growth factor receptor 2 (HER2)-positive, and triple-negative tumours. The current state-of-the art suggests that cannabinoid-based approaches may offer a therapeutic benefit in these three BC subtypes [27].

#### 2.1.1. Cannabinoids and Hormone-Sensitive Breast Cancer

The presence of estrogen receptors (ER) and/or progesterone receptors (PR) in BC cells defines a subgroup of breast tumours that may be susceptible to endocrine therapy. Specifically, patients are treated surgically and/or pharmacologically employing the blockage of estrogenic signalling, which has pro-proliferative features. Targeted strategies either remove the endogenous source of estrogens and/or employ selective ER modulators, such as tamoxifen or inhibitors of aromatase, the main enzyme responsible for estrogen synthesis [24]. It has been demonstrated that cannabinoids modulate pivotal tumour progression-related aspects of ER^+^/PR^+^ BC cells. The endocannabinoid anandamide exerts its anti-proliferative action by blocking the cell cycle progression [28] and by inhibiting adenylyl cyclase thus activating the Raf-1/ERK/MAPK cascade [28,29,30]. This effect is mediated by the activation of CB1R [28,29,30] and is not accompanied by cancer cell death [28]. The proliferation of the ER^-^/PR^+^ human BC cell line EVSA-T also decreased in response to tetrahydrocannabinol (THC) [31,32]. Cannabinoids impair ER^+^/PR^+^ cancer cell migration and invasion in vitro. The selective activation of CB2R in cells over-expressing the chemokine receptor CXCR4 led to the inhibition of chemotaxis and wound healing similar to the effect induced by the CXCR4 ligand CXCL1220 [27].

#### 2.1.2. Cannabinoids and HER2-Positive Breast Cancer

Breast tumours expressing HER2 constitute another breast cancer subtype. HER2 is a member of the epidermal growth factor receptor (EGFR) family, involved in a number of oncogenic processes as cell proliferation and survival [33]. Around 20–30% of primary BC cells exhibit HER2 gene amplification and protein over-expression which is a poor prognostic biomarker and leads to an unfavourable response to chemotherapy [34]. Therapeutic outcomes have improved since the incorporation of Herceptin^®^ (trastuzumab, an antibody against the extracellular domain of HER2) and Tykerb^®^ (lapatinib), a dual EGFR/HER2 tyrosine kinase inhibitor [23,27].

At the same time, CB2R is over-expressed in BC and is present in high levels in aggressive (high-grade) tumours [31,35,36]. The deregulation of the ECS in many cancers has been broadly documented [27,37,38,39], and although there is a strong rationale for using CB2R as an anti-cancer drug target [27,40,41], details on its impact in tumour development and progression are still lacking. Strong preclinical evidence suggest that cannabinoids may be useful for the treatment of this BC subtype. THC exerts a significant anti-tumour action in a model of HER2-positive metastatic BC [42]. THC treatment reduces not only tumour growth, but also the number of generated tumours [36]. Xenograft-based approaches have strengthened the hypothesis that HER2-overexpressing tumours may be sensitive to treatment with THC 14 and/or CB2R-selective agonists [36,43] decreasing tumour growth [43]. Interestingly, the activation of CB2R has been linked with anti-tumour effect of cannabinoids in all HER2-positive BC models used [27].

The protein complexes CXCR4-CB2R, GPR55-CB2R, and HER2-CB2R have been proposed as novel therapeutic targets for HER2+ BC. They possess particular pharmacological and signalling properties, and their modulation might affect the anti-tumoural activity of the ECS. Cannabinoid receptor heteromers have a promising value as new potential targets for BC therapies and as prognostic biomarkers [44,45,46,47].

#### 2.1.3. Cannabinoids and Triple-Negative Breast Cancer

Triple-negative BC lacks the expression of ER, PR and HER2. There is no standard targeted therapy for these patients, whose prognosis is very poor [21]. Attempts have been made to improve chemotherapy responses, like the combined use of angiogenesis inhibitors as Avastin^®^ (bevacizumab) and poly (adenosine diphosphate–ribose) polymerase (PARP) inhibitors [21,23]. Preclinical evidence indicates that this subtype may be treated with cannabinoids. A collection of synthetic cannabinoids have been tested and inhibited cell proliferation [31,35,48,49,50,51,52]. The cannabinoids, via CB1 and/or CB2 receptors, confer a less invasive phenotype to triple negative BC, showing that these compounds may have a reduced cancer cell metastatic potential in vivo [35]. Phytocannabinoids other than THC have also shown anti-tumour actions in BC. Cannabidiol (CBD) has low affinity for CB1R and CB2R [53] and is emerging as an attractive drug in many conditions, although its detailed mechanism of action has still not been elucidated [54,55]. It has been shown that CBD impacts not only proliferation but also metastasis-related capability [27,50,51].

Although there is a high evidence load suggesting the anti-tumour effects of cannabinoids in BC, there have also been reports of their pro-tumourigenic effects [27,56,57,58].

### 2.2. Gastrointestinal Malignancies

Gastrointestinal cancers (GIC) represent a vast family of malignant diseases including rectal cancer, biliary cancer, gastric cancer, esophageal cancer, pancreatic cancer, colorectal cancer, anal cancer, early colon cancer, familial risk colorectal cancer, and hepatocellular carcinoma. Standard treatment approaches depend on various clinical and genetic factors and are constantly evolving to meet the patients’ needs. Despite all invested efforts, colorectal cancer (CRC) is still the third most common malignant disease in the world with around 1.8 million new cases in 2018, and in second place by mortality induced by cancer with around 0.9 million deaths [59]. The ECS’s involvement in the development, progression and treatment of CRC has been evaluated in terms of the implication of cannabinoid receptors, endo- and synthetic cannabinoids, as well as various ECS-induces signalling molecules [60,61].

The expression of CB2R is a poor prognostic factor in CRC and its activation via the AKT/GSK3β signalling pathway has been linked with a more aggressive phenotype [62]. On the other side, the down-regulation of CB1R has been linked with metastatic CRC [63]. Endogenous and synthetic cannabinoids elicit the suppression of CRC cells proliferation and migration and stimulate apoptosis, via receptor-dependent and independent mechanisms [64]. The intracellular pathways include inhibition of RAS–MAPK and PI3K–AKT axis, cell cycle arrest, down-regulation of anti-apoptotic proteins, increased ceramide synthesis, activation of caspases etc. It has previously been shown that traditional phytocannabinoids (THC or CBD) have slightly lower anti-cancer potency in GIC than synthetic compounds, like the CBD derivative HU-331 and CP 55,940 [65,66]. The screening of a library of synthetic cannabinoids led to the discovery of three families of compounds (CP 55,940, CP 47,497, and PTI) able to reduce the viability of CRC cells in vitro [67]. As treatment with antagonists of some CBRs (CB1R, CB2R, GPR55, and TRPV1) did not show a reduction of the activity of these drugs, it was concluded that they might act through a non-canonical receptor mechanism. Modification of these compounds taking into account the different anti-cancer potency of various stereoisomers is suggested as a future direction for the development of novel therapies for CRC. Their use for potentiating the effects of standard chemotherapeutics and in preventing adverse side effects like nausea, vomiting, toxicity, pain and loss of appetite needs to be balanced with their known psychotropic effects [64,68]. The anti-cancer potential of the peroxisome proliferator-activated receptor γ (PPARγ) has also been linked with its affinity towards cannabinoids such as THC and JWH-015 in hepatocellular carcinoma in vitro and in vivo [69]. The up-regulation of *PPARγ* upon cannabinoid binding is considered to have a protective role from inflammation, oxidation, fibrosis, fatty liver and liver tumours [70].

The enzyme monoacylglycerol lipase (MAGL) involved in the metabolism of endogenous cannabinoids is also expressed in higher levels in aggressive CRC cells [71]. Evidence exists that MAGL might modulate angiogenesis, thus its pharmacological inhibition represents a potential new therapeutic approach for the inhibition of CRC progression. The over-activation of the ECS in GIC is associated with poor prognosis and advanced disease stage but reports of its down-regulation in the metastatic setting also exist. This implies that the specific strategy for ECS exploitation in GIC strongly depends on the tumour type and stage.

### 2.3. Gynecological Malignancies

Gynaecological malignancies make up approximately one out of six cancers in women [72]. Although they are usually grouped together, cancers of the female reproductive system comprise a diverse group of cancers with distinct risk factors, signs and symptoms, clinical presentations and treatment modalities, each named after the anatomical part in which the cancer started: cervical, ovarian, uterine (endometrial cancer and uterine sarcoma), vaginal, vulvar, and fallopian tube [73]. Since they play important roles in the regulation of cell proliferation, differentiation and survival, endocannabinoids (ECS) have emerged as a cell regulatory mechanism involved in protection against cancer development. In addition, endocannabinoids are actively involved in all aspects of female reproduction such as oocyte production [74] and their impairment has been associated with various gynaecological pathological conditions such as ectopic pregnancies (N-arachidonoylethanolamine (AEA), CB1R, fatty acid amide hydrolase (FAAH)) and cancer. The expression of CB1R, CB2R, N-acyl phosphatidylethanolamine phospholipase D (NAPE-PLD) and FAAH was shown in normal human ovaries, while AEA was found in the follicular fluid after ovarian stimulation by hormones [75].

#### 2.3.1. Endometrial Cancer (EMC)

CB2 receptors might play a pivotal role in endometrial cancer. It has been shown that CB2R is over-expressed in endometrial cancer biopsies while it’s only weakly expressed in the adjacent normal tissue as well as healthy endometrial tissue [76]. The same study investigated the underlying signalling pathways showing that the complete endogenous pathway of CB2R activation is significantly altered in EMC. They used CB2R over-expressing AN3CA cells to demonstrate a significant reduction in cell vitality compared to parental AN3CA cells. They also showed that incubation with the selective CB2R antagonist SR144128 was able to restore the viability of CB2R over-expressing cells. AN3CA cells transfected with a plasmid containing cDNA for CB2R showed a 40% reduction in mitochondrial function compared to control cells which indicated a potential role of CB2R in the control of EMC cell growth through the modulation of mitochondrial function. Beside CB2R, the endocannabinoid 2-arachidonoylglycerol (2-AG) is present in significantly high levels in cancerous tissues [77]. The significant over-expression of CB2R and 2-AG might be used as a novel therapeutic target for EMC. The expression of CB1R, AEA and palmitoylethanolamine lipid (PEA) were not significantly different between normal and tumour tissue although AEA and PEA showed elevated levels in EMC [76]. Statistical significance was reached in the study by Ayakanny et al. who demonstrated that plasma and tissue AEA and PEA levels were significantly higher in EMC than in controls [78]. Since their levels are significantly higher in plasma of EMC patients than in the healthy controls, these biomarkers can be useful in early and non-invasive diagnosis of endometrial cancer.

#### 2.3.2. Ovarian Cancer (OC)

It was shown that aggressive ovarian cancer cells (SKOV3) display significantly elevated MAGL hydrolytic activity compared to non-aggressive cells (OVCAR3). MAGL degrades 2-AG which is also found in elevated levels in high-grade primary human ovarian tumours [79]. Induced over-expression of MAGL in non-aggressive cancer cells increases their pathogenicity. This effect is reversed by MAGL inhibitors which have an important therapeutic potential. GPR55 has also been investigated in OC cell lines. High GPR55 expression on both protein and mRNA levels was shown in OVCAR2 and A2780 OC cell lines [80]. CB1R expression was moderate in benign and borderline epithelial ovarian tumours but it was strongly increased in invasive ovarian tumours [81].

#### 2.3.3. Cervical Cancer (CC)

As conventional chemotherapy has limited success in the reduction of cervical cancer (CC) mortality, the influence of various plant-derived products in the development and treatment of this disease has been investigated in recent years [82,83,84]. While investigating Cannabis sativa and the ECS in this setting, a specific expression pattern of CB1R, CB2R, and TRPV1 in CC cell lines and tumour biopsies was observed [85]. The investigation of the effect of AEA on CC cell lines also showed interesting results. HeLa and CC299 cells were sensitive to AEA that induced DNA fragmentation leading to cell cycle arrest and cell death. Interestingly, selective CB1R and CB2R antagonists enhanced the toxic effects of AEA suggesting possible protective effect of CB1R and CB2R in AEA-induced cell death [85]. Contrary to this, TRPV1-selective antagonist capsazepine (CZ) protected cells against AEA, suggesting that TRPV1 is involved in the mechanism of AEA-induced apoptosis in cervical cancer cell lines. In the CC cell lines HeLa and C33A, CBD was able to decrease the invasiveness in a concentration-dependent manner by the up-regulation of TIMP metallopeptidase inhibitor 1 (TIMP-1) via CB1R/CB2R and TRPV1 [86]. Also, CBD-induced cell death by accumulation of cells in the sub-G0 phase which was most likely related to caspase-9 and caspase-3 up-regulation upon CBD treatment [82]. Based on these in vitro studies, in vivo studies should be initiated to investigate CBD as an additional therapeutic tool in cervical cancer treatment.

### 2.4. Prostate Cancer (PC)

Prostate cancer is one of the most common malignant cancers in men. It is the second most frequently diagnosed cancer and one of the leading causes of cancer death worldwide in the male population [87]. Standard treatment of localized PC is surgery or radiotherapy. Approximately one third of conventionally treated patients will develop metastases, at which point androgen withdrawal is the most effective form of systemic therapy. Unfortunately, androgen deprivation is associated with a gradual transition of PC cells through a spectrum of androgen dependence, followed by androgen sensitivity, and finally androgen independence, known as castration-resistant prostate cancer (CRPC). This stage of PC has a more aggressive phenotype and is unresponsive to further hormonal therapy, with a very poor prognosis [88,89]. Cannabinoids have shown a high anticancer activity in PC, but the specific molecular mechanisms responsible for these effects depend on the drug and tumour context.

In PC, abnormal expression of ECS has been found. This has been related to cancer prognosis, suggesting a potential therapeutic implication of ECS in tumour progression. Anandamide levels are elevated more than threefold [77], and CB1R and CB2R expressions are also increased in PC [90,91]. High expression of CB1R has been associated with poor prognosis. In vitro data also showed that GPR55 is expressed in LNCaP, PC3, and DU145 prostate cancer cell lines, where it signals via calcium mobilization and the activation of Akt and ERK1/2 [80].

Phytocannabinoids, endocannabinoids, and synthetic cannabinoids have proved to inhibit prostate tumour cell proliferation, migration, and metastasis, as well as to induce apoptosis. Various authors have shown endogenous 2-AG as a potential inhibitor of androgen-independent prostate cancer cells invasion [92], by inhibiting adenylyl cyclase and reducing protein kinase A (PKA) activity, suggesting that these effects are mediated by CB1R [92,93,94]. Noladin ether has also proven to inhibit the invasion of PC cells [95]. An increase in endogenous 2-AG levels after MAGL inhibition has also been shown to interfere with cancer progression. MAGL inhibitors lower the invasive capacity of PC and this effect is partially reversed by the blockage of CB1R [93,94,96]. The disruption of MAGL activity lowers EGFR expression, thus reducing the EGF-induced cell proliferation [97]. Sundry’s studies have evidenced the anti-proliferative activity of cannabinoids in prostate tumours. Anandamide inhibits the proliferation of cells (PC-3, DU-145, and LNCaP) [98,99] and primary cultures of PC [91] via CB1R. Phytocannabinoids also reduce PC cell proliferation. The two main cannabinoids from the marijuana plant (delta-9-tetrahydrocannabinol (Δ9-THC) and CBD) cause cell death in PC-3 and LNCaP PC cell lines, respectively, inducing apoptosis in LNCaP cells [100,101]. However, the anti-proliferative activity of CBD and Δ9-THC does not involve cannabinoid receptors. Other synthetic cannabinoids, such as WIN-55,512-22, JWH-015, and HU‑210 also exert antitumour effects in PC cells, as they inhibit cellular proliferation in androgen-insensitive PC cell lines [30,90,93,94,102,103].

It has recently been reported that CB2R can form heteromers with the GPC chemokine receptor CXCR4 in PC cells [44,47,104]. CXCR4 is involved in various mechanisms that enhance the cell’s ability to proliferate and migrate, thus its activation has been linked to local and distant metastatic invasion. This heteromerization might enable cannabinoids to indirectly reduce the invasive properties of cancer cells by inhibiting the effects of CXCR4 agonists [44,47,104]. These data point to a novel pharmacologic target affecting tumour cell migration and invasion that could be useful in the metastatic setting.

### 2.5. Thoracic Tumours

In 2018, over two million new lung cancer (LC) cases were diagnosed, and over 1.3 million people have died from LC, making this disease the most common occurring malignant disease in the world, as well as the most common cause of cancer-related deaths [59]. Although LC is a model cancer for the success of molecular targeted therapies [105,106], due to the high cost of radiologically-based nation-wide screening programs [107,108], it is most often diagnosed in advanced disease stages when the level of cancer-related pain is high [109]. An individual combination of pharmacological and non-pharmacological approaches for each patient ensures the optimal palliative care which results in higher quality of life and longer survival. The role of the ECS is ambiguous in LC, as there have been sporadic reports connecting the use of cannabinoids to a higher risk of LC [110] and more reports that document its beneficiary properties. Although it is known that cannabis contains many similar toxins and carcinogens to tobacco [111] and regular marijuana use has been shown to induce various pulmonary problems [112,113], to date, there are no conclusive data associating it with an increased risk of lung cancer [114,115].

Most reports on this subject have dealt with the benefits of cannabinoids in the control of LC-induced pain and therapy side-effects [116]. However, the burden of evidence on the efficacy of concurrent cannabis use with various cancer treatments is still not sufficiently strong to result in official recommendations of their use in this setting. The interaction between the downstream effects of approved chemo-, targeted and immunotherapy drugs for LC [106] and the metabolism of cannabinoids is complex, which calls for caution in the interpretation of data derived from uncontrolled studies. Some ECS components have shown a direct anti-cancer potential by modulating various signalling pathways (ERK, PI3K, p38 MAPK, ceramide pathways), thus inducing apoptosis and/or the inhibition of cell proliferation and epithelial-to-mesenchymal transition (EMT) [117]. Cannabinoid receptors CB1R and CB2R have been shown to be over-expressed in LC at the genetic level, and this was associated with prolonged survival of patients [117]. The agonist of CB2R JWH-015 was assessed in an in vivo tumourigenesis model and had the ability to inhibit the EMT process of LC cells by down-regulating *EGFR* signalling which is usually markedly increased in LC [118]. THC and CBD also suppressed the basal EMT phenotype in vitro, by down-regulation of cadherin 1 (*CDH1*) and up-regulation of cadherin 2 (*CDH2*) and vimentin (*VIM*). Synthetic cannabinoids WIN-55,212-2 and JWH-133 have also been linked with the inhibition of growth and metastasis of LC cells in vitro and in vivo by blockage of Akt phosphorylation and lowering the levels of matrix metalloproteinase-9 (MMP9) [119]. CBD-induced effects in LC cells have also been shown to be non-canonical, inducing the expression of PPAR-γ and cyclooxygenase-2 (COX-2) [120]. PPAR*γ* is a ligand-activated transcription factor that may function as a tumour suppressor upon stimulation with cannabinoids in LC cells, through its ability to regulate angiogenesis and production of matrix metalloproteinases in the LC microenvironment [121]. Activation of cannabinoid receptors can also selectively inhibit the lung-resident macrophages-induced release of angiogenic stimulators, thus modulating the complex process of vascular remodelling crucial for cancer growth and inflammation [122].

### 2.6. Thyroid Cancer (TC)

Fewer than 1% of all thyroid nodules are cancerous and, even when they are, most of thyroid cancers are very curable. In fact, the most common types of TC (papillary ~85%, follicular ~10%) are most curable in patients younger than 50, with a 98% cure rate if treated appropriately. On the other hand, there are rare forms such as anaplastic TC which are very aggressive (median survival 3–5 months) [123]. Even though these types of cancer are very rare (less than 2% of all thyroid cancers) therapeutic options are needed for these aggressive forms of disease.

Even though there is a limited number of studies that investigated the effect of cannabinoids on thyroid tumour development in vivo, some of them showed ECS’ involvement in tumour growth modulation. It was reported that stimulation of CB1R by the endocannabinoid analog 2-methyl-arachidonyl-2′-fluoro-ethylamide (Met-F-AEA) inhibits the growth of rat TC cell-derived tumour in athymic mice by inhibiting p21ras activity [124]. In addition, it was shown that Met-F-AEA is also able to block the growth of already established tumours by inhibiting the expression of vascular endothelial growth factor (VEGF) [125]. Since VEGF upregulation has been associated with malignancy in human thyroid tumours and cancer cells [126] it is important to note that anandamide-based drugs may be efficacious therapeutic drugs for the inhibition of cancer cell growth. Assuming that the substances that inhibit the degradation of endocannabinoids should also be capable of inhibiting cancer growth in vivo, the effect of endocannabinoid degradation inhibitors on the growth of rat thyroid tumour xenografts induced in athymic mice was investigated [127]. It was shown that agents that inhibited EMT (VDM-11) and blocked AEA hydrolysis (AA-5HT) prevented in vivo tumour growth. Similarly, the endocannabinoid 2-AG reduced thyroid tumour development.

In the study by Shi et al., the synthetic cannabinoid JWH-133 was tested in the highly aggressive anaplastic TC cell line ARO tumour model [128]. They investigated gene expression profiles of ARO and ARO-IL/12 (cell line with lower tumourogenicity after interleukin (IL)-12 gene transfer) by microarray analysis of 3757 genes. CB2R gene (CNR2) was expressed eightfold higher in ARO/IL-12 cells than ARO cells and at the same time was the most highly expressed gene in these experiments. This was the study that demonstrated for the first time that CB2R expression is induced following IL-12 expression in ARO cell line. The over-expression of CB2R makes cells more susceptibile to CB2R agonist-mediated apoptosis and regression of tumours. Based on this assumption they further showed that CB2R agonist JWH133 and mixed CB1R/CB2R agonist could induce a significantly higher rate of apoptosis in ARO/IL-12 than ARO cells. Local administration of JWH133 showed a considerable regression of thyroid tumours in nude mice generated by inoculation of ARO/CB2R cells. Furthermore they demonstrated a significant increase in apoptosis in ARO/IL12 and ARO/CB2R cells following incubation with 15 nM paclitaxel which showed senzitization of tumour cells to chemotherapy.

The results of these studies suggest that manipulation of the ECS can be consider as an option to prevent propagation of thyroid tumour cells and that CB2R may be a therapeutic target for the treatment of the most aggressive types of TC. We note that in vivo TC studies with cannabinoids are scarce and more rigorous evaluation is needed to confirm the role of the ECS in this malignancy.

### 2.7. Central Nervous System Malignancies

There are over 130 types of brain tumours, as classified by the World Health Organisation. Brain tumours can differ in the cells they originate from, how quickly they are likely to grow and spread, and the location of the brain they affect. The most common types of adult brain tumours or gliomas include glioblastoma, astrocytoma, meningioma and pituitary adenoma. Gliomas are defined as the tumours that display histological, immunohistochemical, and ultrastructural evidence of glial differentiation. They are classified according to cellular features and grade of malignancy [129]. Glioblastoma multiforme (GBM), or grade IV astrocytoma, is the most frequent class of malignant primary brain tumours being the most aggressive form of cancer. Consequently, survival after diagnosis is low [129,130], due primarily to the high invasiveness and proliferation rate of GBM. Additionally, GBM exhibits a high resistance to standard chemotherapy and radiotherapy. Current standard therapeutic strategies for the treatment of GBM are only palliative including surgical resection and focal radiotherapy [130,131]. It has been recently found that cannabinoids exert anti-glioma actions in laboratory animals and constitute a potential cannabinoid-based therapy for GBM [132].

Most of our research on cannabinoid anti-tumoural action has focused on gliomas [133]. Glioma cells have been used as the most common model system for studying cannabinoid-induced anticancer mechanisms. Initial studies showed that cannabinoids can induce apoptosis of glioma cells via CB1R and CB2R dependent de novo synthesis of the sphingolipid ceramide showing pro-apoptotic properties [47,94,134,135,136,137]. CB1R is over-expressed in glioblastomas [138] and paediatric low-grade gliomas, and is implicated in tumour involution induced by apoptosis and cell-cycle arrest upon activation [139]. CB_2_R is also highly expressed in glioblastomas and astrocytomas and related to tumour grade [94,137,138,140,141]. While some authors have observed that AEA levels are lower in gliomas, compared with non-tumour tissue [138,142], others have detected higher levels of this endocannabinoid in gliomas and also in meningiomas [143]. Regarding 2-AG level, it was up-regulated in both brain tumour types [138,143]. Various authors have shown that AEA inhibited in vitro proliferation of several glioma cells via induction of apoptosis [85,94,144,145]. It also decreased the migration and invasion of these cells [146,147]. In addition to AEA, 2-AG and other endocannabinoids reduced the proliferation of C6 glioma cells [148] and these effects were mediated by CBRs [149]. Cannabidiol and Δ9-THC, administered alone or in combination, have also displayed an anti-proliferative effect on several glioma cell lines, inducing apoptosis, with the participation of CB2R [94,150,151].

Animal model studies have shown that local administration of THC or WIN-55,212-2 reduced the tumours formed by intracranial inoculation of C6 glioma cells. This led to eradication of gliomas and increased survival in one third of treated rats [132,134]. Local administration of THC, WIN-55,212-2, or JWH-133 also slowed down tumour growth derived from rat glioma C6 cells and GBM cells obtained from patient tumour biopsies [132,134,137]. These and other studies also showed that activation of cannabinoid receptors on glioma cells modulates important signalling pathways involved in cell proliferation and survival. The downstream anti-cancer cannabinoid-induced events in gliomas have not been elucidated in detail, but there is substantial evidence to confirm their role in apoptosis and inhibition of angiogenesis [132]. Finally, cannabinoids have shown anti-tumour activity in brain cancer.

One of the first studies performed to evaluate cannabinoids’ antitumoural actions was performed by Guzmán and collaborators, who showed that cannabinoids can inhibit tumour growth [47,133]. Due to ethical and legal issues, the first studies were conducted in terminal patients with recurrent tumours [47,152]. These studies elaborated on their palliative effects, but also on their potential anti-cancer effects, alone or in combination with other drugs. In 2017, a phase II, randomized, placebo-controlled clinical trial with recurrent GBM patients was announced and showed the potential efficacy of cannabinoids as add-on anticancer drugs. A combination of THC and CBD in addition to dose-intensive temozolomide was tested. This study showed a significantly higher one-year survival rate in the cannabinoid-treated group (83% vs. 53%), and the median survival was also longer (550 days compared to 369 days) (GW Pharmaceuticals, 2017 press release; ClinicalTrials.gov Identifiers: NCT01812616, NCT01812603).

### 2.8. Melanoma

Melanoma represents an aggressive form of malignant skin cancer which develops by transformation of melanocytes. Despite the introduction of targeted therapies and immunotherapy for the treatment of malignant melanoma, it is still associated with significant morbidity and mortality [59]. In order to improve the prognosis of these patients, repurposing of already approved drugs for other uses has been suggested as a viable approach [153] as well as re-evaluation of targets, like the skin ECS, that have shown benefit in other conditions and uses.

In a recent study, a treatment of mice with CBD induced a significant decrease in tumour size compared to placebo, and an increased survival and movement ability was also detected [154]. The activation of cannabinoid receptors on melanoma cells can lead to G1-cell cycle arrest by the inhibition of Akt and pRb signalling molecules, activation of caspase-3, stimulation of ROS production, and inhibition of the expression of EGF and VEGF, which in turn lowers the proliferation and metastatic potential of melanoma cells [155]. CB2Rs is over-expressed in melanoma [156]. However, the complex interactions between the inflammatory component present in the skin tumour microenvironment and the ECS can lead to various outcomes depending on the level of ECS activation and the specific ECS component. The activation of CB2R by CBD can lead to anti-inflammatory and immuno-modulatory effects, which in turn might regulate the overall response of melanoma cells to therapy [154]. AEA, THC and synthetic cannabinoids WIN-55,212–2 and JWH-133 have also shown some anti-cancer potential, acting through CB1R and CB2R [157,158]. The application of the endocannabinoids AEA, 2-AG, as well as the endogenous signalling lipid PEA and inhibitor of FAAH involved in ECS metabolism were shown to increase cell death both in vitro and in vivo [159]. On the contrary, reports of a pro-tumourigenic effect of CB1R also exist [160]. This further emphasizes the interplay between the ECS, the specific cancer cell type and the immune microenvironment which needs to be considered when designing future studies. The dose of the applied cannabinoid, as well as its complex interaction with the primary anti-cancer therapy regimen via intersecting downstream signalling pathways might have a significant impact on the final outcome [161,162].

## 3. Legal and Ethical Aspects of ECS Exploitation in Oncology

While clinical trials employing phytocannabinoids as CBD or targeting other components of the ECS in cancer pose no more ethical issues than the ones that appear in almost every human-related oncological clinical trial [163], medical, ethical and legal ramifications of the use of exogenous psychotropic cannabinoids as THC are vast. Beside the favourable benefit-to-risk ratio, fully autonomous and informed consent and careful monitoring for safety and side effects, additional ethical considerations related to social context and lingering misconceptions are related to medicinal cannabis use.

Cannabinoids have an important role in palliative medicine due to their analgesic and antiemetic effects, but an increasing number of preclinical studies indicate their anticancer properties as well. Even though some cannabinoid-based drugs have been registered in several countries (e.g., nabiximols, dronabinol, nabilone), there have been studies demonstrating moderate- or low-quality evidence supporting the use of these agents in anti-cancer treatment [164]. The ethical and medical debates are still ongoing about the use of psychotropic cannabinoids as therapeutics in cancer patients. The proof of profound safety and efficacy in clinical trials is lacking and it is hard to assess the potential benefits and risks. Many aspects are still unknown about the way of administration, dosage, interaction with other drugs and adverse effects. The legal prohibition of medical marijuana on the other hand directly confronts the personal and autonomous freedom of choice. It might be said that medical facts are still too vague to overturn the informed decision that harms are not inflicted to third parties when marijuana is used for medicinal purpose and that possible harms cannot outweigh the suffering that can probably be removed by the drugs [165]. The social and political history of cannabis prohibition and the stigma it has perpetuated continues to stand in the way of detailed systematic research that will help elucidate many dilemmas pertaining to its use. To help guide the research in this exciting medical filed, the principles of biomedical ethics, i.e., respect for autonomy, beneficence, and justice, should be followed.

## 4. Conclusions

The use of ECS components as anti-cancer agents and targets, and the range of effects they might induce (cell death, regulation of angiogenesis and invasion or anticancer immunity), depend in great deal on the specific cannabinoid ligand acting in a specific cancer cell type. Although an attractive target, the use of ECS components in anti-cancer treatment is interlinked with many legal and ethical issues that need to be considered. The legislation which outlines the permissive boundaries of their therapeutic use in oncology is still unable to follow the current scientific burden of evidence, but the number of ongoing clinical trials might tip the scale forward in the near future.

## Figures and Tables

**Figure 1 cancers-12-03275-f001:**
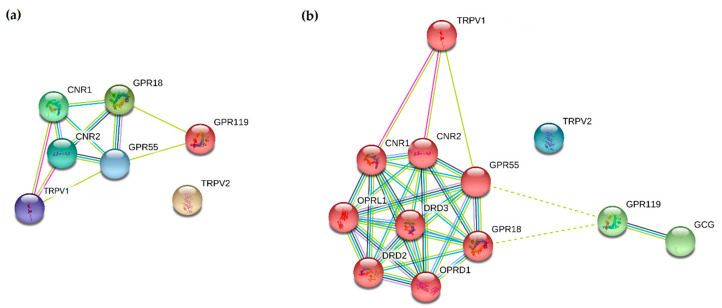
STRING interaction analysis of cannabinoid receptors. (**a**) Direct STRING analysis network was built based on high confidence (0.7) evidence from experimental interaction data (pink), co-expression (black), gene neighbourhood (green) and co-occurrence (blue) data, curated databases (light blue), protein homology (purple), and predictive and knowledge text mining (light green); (**b**) The network was extended to 5 primary-interaction shell genes to explore their indirect interactions and clustering (PPI enrichment *p*-value: 2.39 × 10^−12^) using the intersection of 12 genes present on all analysed platforms. Red nodes—*CNR1/CNR2* cluster, green nodes—GPR119 cluster, blue nodes—TRPV2 cluster. Nodes are labelled with Human Gene Nomenclature Committee (HGNC) gene symbols: *CNR1*—Cannabinoid receptor 1 gene, *CNR2*—Cannabinoid receptor 2 gene, DRD2—dopamine receptor D2, DRD3—dopamine receptor D3, GCG—glucagon, GPR18—N-arachidonyl glycine receptor (G-protein coupled receptor 18), GPR55—G-protein coupled receptor 55, GPR119—Glucose-dependent insulinotropic receptor (G protein-coupled receptor 119), OPRD1—Opioid Receptor Delta 1, OPRL1—Opioid Related Nociceptin Receptor 1, TRPV1 and TRPV2—transient receptor potential cation channel subfamily V members 1 and 2.

**Figure 2 cancers-12-03275-f002:**
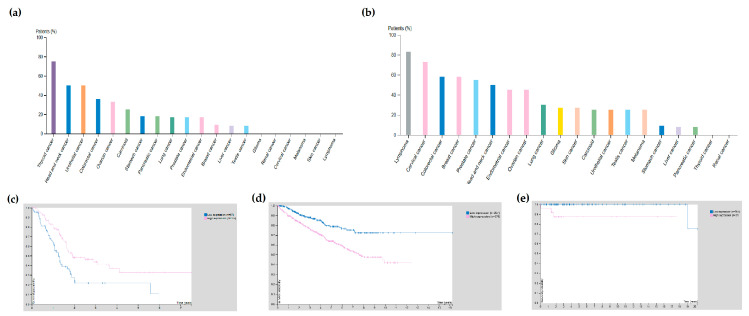
The expression of *CNR1*, *CNR2*, *GPR119* and *TRPV2* in cancer according to the Human Protein Atlas database [13]. (**a**) Expression of *CNR1* in cancer subtypes; (**b**) expression of *CNR2* in cancer subtypes; (**c**) survival curves of pancreatic cancer patients according to the expression of *GPR119* favourable prognostic factor, *p* < 0.001); (**d**) survival curves of renal cancer patients according to the expression of *TRPV2* (unfavourable prognostic factor, *p* < 0.001); (**e**) survival curves of testicular cancer according to the expression of *TRPV2* (unfavourable prognostic factor, *p* < 0.001).

**Table 1 cancers-12-03275-t001:** Comparison of cannabinoid receptors’ genetic expression in normal vs. tumour tissue according to the UALCAN database [16] analysis of TCGA data.

Cancer Type	Receptor	Cancer Subtype	Expression in Tumour vs. Normal Tissue	*p* Value
Breast cancer	CNR1	Breast invasive carcinoma	Under-expressed	7.09 × 10^−11^
	CNR2	Breast invasive carcinoma	Under-expressed	1.55 × 10^−2^
	GPR18	Breast invasive carcinoma	Over-expressed	3.60 × 10^−7^
Gastrointestinal malignancies	CNR1	Cholangiocarcinoma	Over-expressed	3.16 × 10^−2^
		Colon adenocarcinoma	Under-expressed	1.58 × 10^−7^
		Hepatocellular carcinoma	Over-expressed	3.52 × 10^−11^
		Rectum adenocarcinoma	Under-expressed	9.80 × 10^−3^
	CNR2	Colon adenocarcinoma	Under-expressed	6.57 × 10^−4^
		Rectum adenocarcinoma	Under-expressed	2.83 × 10^−2^
	GPR18	Colon adenocarcinoma	Under-expressed	3.30 × 10^−6^
	GPR55	Colon adenocarcinoma	Under-expressed	2.16 × 10^−4^
	GPR119	Colon adenocarcinoma	Under-expressed	1.55 × 10^−5^
		Hepatocellular carcinoma	Under-expressed	3.58 × 10^−5^
		Pancreatic adenocarcinoma	Over-expressed	1.73 × 10^−2^
		Rectum adenocarcinoma	Under-expressed	2.84 × 10^−3^
	TRPV1	Hepatocellular carcinoma	Over-expressed	4.75 × 10^−6^
		Stomach adenocarcinoma	Over-expressed	1.32 × 10^−3^
	TRPV2	Cholangiocarcinoma	Over-expressed	5.71 × 10^−7^
		Hepatocellular carcinoma	Over-expressed	4.27 × 10^−9^
		Stomach adenocarcinoma	Over-expressed	1.22 × 10^−8^
Gynaecological malignancies	CNR1	Uterine corpus endometrial carcinoma	Under-expressed	1.54 × 10^−2^
	GPR18	Cervical squamous cell carcinoma	Over-expressed	1.18 × 10^−3^
	GPR55	Cervical squamous cell carcinoma	Over-expressed	1.64 × 10^−9^
		Uterine corpus endometrial carcinoma	Under-expressed	9.88 × 10^−7^
Prostate cancer	CNR1	Prostate adenocarcinoma	Under-expressed	3.45 × 10^−6^
	TRPV1	Prostate adenocarcinoma	Over-expressed	1.05 × 10^−4^
	TRPV2	Prostate adenocarcinoma	Under-expressed	3.56 × 10^−2^
Thoracic tumours	CNR1	Lung adenocarcinoma	Under-expressed	1.62 × 10^−12^
		Lung squamocellular carcinoma	Under-expressed	4.06 × 10^−7^
	TRPV1	Lung adenocarcinoma	Over-expressed	<1 × 10^−12^
		Lung squamous cell carcinoma	Over-expressed	6.11 × 10^−10^
	TRPV2	Lung adenocarcinoma	Under-expressed	<1 × 10^−12^
		Lung squamous cell carcinoma	Under-expressed	1.62 × 10^−12^
Thyroid cancer	CNR1	Thyroid carcinoma	Under-expressed	3.05 × 10^−2^
	CNR2	Thyroid carcinoma	Under-expressed	1.72 × 10^−5^
	GPR18	Thyroid carcinoma	Under-expressed	3.94 × 10^−3^
	GPR55	Thyroid carcinoma	Over-expressed	2.24 × 10^−4^
	TRPV1	Thyroid carcinoma	Under-expressed	3.74 × 10^−2^
Central nervous system malignancies	GPR18	Glioblastoma multiforme	Over-expressed	1.60 × 10^−6^
	TRPV1	Glioblastoma multiforme	Under-expressed	4.78 × 10^−2^
Melanoma (primary vs. metastasis)	CNR2	Skin cutaneous melanoma	Over-expressed	1.22 × 10^−6^
	GPR18	Skin cutaneous melanoma	Over-expressed	1.61 × 10^−9^
	GPR119	Skin cutaneous melanoma	Under-expressed	1.95 × 10^−2^
	TRPV2	Skin cutaneous melanoma	Under-expressed	3.81 × 10^−2^

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
