# Peer review of "The Interplay between Cancer Biology and the Endocannabinoid System—Significance for Cancer Risk, Prognosis and Response to Treatment"

_cancers, 2020, doi:10.3390/cancers12113275_

Round 1
Reviewer 1 Report
The review entitled “The interplay between cancer biology and the endocannabinoid system – significance for cancer risk, prognosis and response to treatment” and written by Moreno E and colleagues presents the current knowledge regarding endocannabinoid system in relation to several types of solid cancers.
I find the present article to be very informative, detailed, well written and with extremely important content. It deserves publication.
However, I have some suggestion that might improve the present manuscript:
- Please define the acronyms the first time you use them in the text
- A list of abbreviations in the end of the article will be beneficial to the readers as there are many abbreviations in the manuscript
- Please read again all the manuscript as English language and spell check are required
- Line 509 please include “as” after ECS components
Author Response
We thank the reviewer 1 for her/his constructive comments.
We have defined all the acronyms and abbreviations and have included the list of abbreviations at the end of the article. We have added “as” after ECS components in the conclusion section (line 509 of the first Manuscript version) where it was missing.
We have read again the whole manuscript and spell-checked it. We consulted a native English speaker who performed the necessary English language check and editing.
Reviewer 2 Report
The authors present here a comprehensive review of the involvement of endocannabinoid system in cancer and highlighted potential therapeutic avenues. The review was very well written, thoroughly researched with appropriate references.
I believe this review would be a wonderful resource for the researchers in this field and I would recommend the manuscript to be published as currently presented.
Author Response
We thank the reviewer 2 for her/his positive comments.
Reviewer 3 Report
This is a particularly well-written review, summarizing the most important results concerning the roles of cannabinoids and expression of different components of cannabinoid system in different cancers. I particularly like the fact that authors used the freely available data from open-access databases like Human Protein Atlas and UALCAN to underlie their points.
Thus, my recommendation will be to accept the manuscript in present form.
Author Response
We thank the reviewer 3 for her/his positive comments.